# Patients’, Pharmacy Staff Members’, and Pharmacy Researchers’ Perceptions of Central Elements in Prescription Encounters at the Pharmacy Counter

**DOI:** 10.3390/pharmacy7030084

**Published:** 2019-07-04

**Authors:** Susanne Kaae, Lotte Stig Nørgaard, Sofia Kälvemark Sporrong, Anna Birna Almarsdottir, Mette Kofoed, Rami Faris Daysh, Nima Jowkar

**Affiliations:** 1Social and Clinical Pharmacy, Department of Pharmacy, Faculty of Health and Medical Sciences, University of Denmark, 2100 København Ø, Denmark; 2Department of Public Health, University of Southern Denmark, 5000 Odense C, Denmark

**Keywords:** pharmacy communication, cue orientation, focus group interviews, Denmark

## Abstract

**Background**: Studies suggest that the way pharmacy counselling takes place does not fully support patients in obtaining optimal medicine use. To understand the basis of current challenges in pharmacy counselling, we investigated which selected related cues, i.e., objects, sounds, or circumstances in prescription encounters, patients, and pharmacy staff notice, and how they interpret these cues. Pharmacy practice researchers’ cue orientation was also investigated to explore possible differences to those of staff and patients. **Methods**: Twelve focus group interviews representing 5 community pharmacies (staff and patients) and 2 universities (researchers) were conducted during 2017–2018 in Denmark. A total of 20 patients, 22 pharmacy staff, and 6 pharmacy researchers participated. A theoretical analysis based on cue orientation and social appraisal was conducted. **Results:** Pharmacy staff, patients and researchers noticed different selected related cues in prescription encounters. Staff particularly noticed ‘types of patients’. Patients were more divided and grouped into three overall categories: ‘types of staff’, medical content, and the situation around the encounter. Pharmacy researchers noticed multiple cues. Different emotions were integrated in the construction of the cues. **Conclusion:** Differences in the cue orientation between all three groups were identified. The identified types of cues and emotions can explain an underlying dissatisfaction with the encounters. Patients lack, in particular, more personal contact. Staff need to consider these aspects to provide relevant counselling.

## 1. Introduction

Communication between pharmacy staff and patients at the pharmacy counter is important for community pharmacies to fulfil their societal obligations of offering professional counselling to patients. Medication counselling is a challenging process that should take the wishes and needs of the patient into consideration, while at the same time performing quality dispensing and keeping waiting times to a minimum.

Several aspects of medication counselling have been investigated, especially those regarding how often staff provide information, what the information concerns, the duration of the encounters, the quality of the counselling, and staff and patient characteristics with an influence on the communication [1,2,3,4,5,6,7,8]. Studies suggest that the way pharmacy counselling takes place today does not fully support patients in obtaining optimal medicine use. Identified problems include patient reliance on doctors rather than pharmacy staff to provide relevant information [9], that staff do not activate patients in the encounters [10,11,12,13], and that patients do not always find information from the pharmacy relevant [14]. However, the underlying reasons for the identified challenges remain unknown.

One important aspect in improving pharmacy counselling is to understand how counselling is perceived by pharmacy staff and patients since the ways they approach and react to the encounters are shaped by these perceptions. In any face-to-face encounter, there are, according to classic communication theory, potentially many elements/cues i.e., objects, sounds, or circumstances to be taken into consideration by participants in order for them to interpret what is taking place [15]. Cues in a pharmacy could be the architecture, characteristics of the person at the counter, etc. Which cues are selected for interpretation and how they are interpreted differ from preconceptions based on cultural learning and personal experiences [15]. Hence, examining what cues pharmacy staff and patients notice in pharmacy encounters, and which meanings and values they ascribe to them, will help us understand their basic understanding of pharmacy encounters. The underlying perceptions of pharmacy encounters could give valuable insight into why counselling challenges exist, especially in the case of discrepancies between patients and staff.

One might assume that pharmacy practice researchers also develop certain perceptions of pharmacy encounters that influence their research, both in terms of perspectives and interpretations. It is important to find out what these are, especially since they may differ from those of both community pharmacy staff and patients.

The aim of this study was, therefore, to investigate what cues in community pharmacy encounters patients, staff and pharmacy practice researchers notice, and how they interpret these cues. 

Due to identified differences in pharmacy patients’ interest in receiving over-the-counter versus prescription counselling [14,16], the study was restricted to prescription encounters. This choice was also made because prescription encounters, especially those involving refill prescriptions, appear to be particularly challenging in terms of counselling [14,16].

## 2. Materials and Methods

To investigate pharmacy patients’, staff members’ and researchers’ cue orientation in prescription encounters, focus group interviews were conducted. This method is ideal for stimulating discussion, gaining insights and generating ideas of social issues under investigation [17].

### 2.1. Theoretical Framework

Only a small fraction of data from our surroundings is ever consciously perceived in any way by an individual. Of this fraction, only a selected portion (cues) is in some sense chosen by the individual. We further simplify the waves of incoming data by recording their contents into meaningful ‘*summary codes*’. The overall meaning we make of selected and usually related cues is influenced by our ‘expectancy set’, which is described as: “One’s cultural belief system learned through socialization, the sum of one’s experiences, and one’s currently salient roles” [15]. With regard to the selection of incoming data, individuals tend to focus on cues that reinforce past or emerging interpretations [18].

In this study, the articulated focus of incoming data is defined as ‘selected related cues’ i.e., of all the possible impressions in a pharmacy prescription encounter, what types of cues do the different actors overall notice? The summary and interpretation of the ‘selected related cues’ i.e., which elements do participants include in their description of ‘selected related cues’, are defined as ‘summary codes’. Patients, pharmacy staff and researchers have individual knowledge and experiences but are expected to share some cultural belief systems (in particular staff and researchers through education and workplace socialization), which might create different patterns of cue orientation and summary codes between the groups.

### 2.2. Design

To investigate selected related cues and summary codes, an exploratory study design was chosen to cover participants’ untainted perceptions of and experiences with prescription encounters. The interview guide was thus kept short and open and consisted of only two major questions. The first question was presented in three ways to stimulate as many reflections as possible:-What types of encounters have you experienced at the pharmacy counter?-What types of meetings have you experienced at the pharmacy counter?-Which types of different (human) interactions have you experienced at the pharmacy counter?

The second question concerned participants’ opinions about the role of community pharmacies in society, this was to explore if and how these views might influence the way participants react in the pharmacy encounters.

To identify participants’ original cue orientation, they were first asked to write their immediate thoughts about the first question for 5–10 minutes, and each participant was then asked to tell the other participants about their notes. Participants then commented on each other’s remarks to generate more reflections. By this design, both the individuals’ untainted cue orientation was identified along with the benefit of the focus group participants discussing the cues with each other. By the end of the interview, the second question was presented. Data collectors used probing questions when necessary. The three types of interviewees were interviewed in separate focus groups to explore their perceptions in depth. The interviews were recorded and transcribed verbatim.

### 2.3. Recruitment

Five community pharmacies in Denmark were approached for purposeful yet convenient sampling ensuring heterogeneity between the pharmacies according to location (provincial/urban) and overall socio-demographic background of inhabitants in the area. All agreed to participate. The pharmacies were asked to recruit 4–6 staff members (both pharmacy technicians and pharmacists) and 4–6 patients from the pharmacy for the interviews. Inclusion criteria for patients were adulthood, recipient of prescription medicine and a variation in gender and age. Two groups of researchers from the University of Copenhagen and from the University of Southern Denmark both engaged in pharmacy practice research represented the views of pharmacy researchers.

The interviews took place at the five pharmacies and at the two universities. Four of the authors (first, fifth, sixth and last author) conducted the interviews.

### 2.4. Analysis

Transcripts were read, and relevant citations were extracted and coded in NVivo12. Codes were divided into whether the findings were from staff, patients or researchers. The codes and extracts were theoretically interpreted according to the type of *selected related cues* (what items were mentioned when participants were speaking of pharmacy encounters) and *summary code* (what different elements were described by participants in relation to the selected related cues) [19]. All authors, individually, carried out the initial stage of the analysis for three pharmacies (patient and staff focus groups). Selected related cues and summary codes identified were then compared in consensus discussions. Based on these discussions the first author undertook the analysis of the pharmacy researchers. This was as three of the authors participated as interviewees in one university interview, and hence were not suited for this.

Interviews at pharmacies (pharmacy staff and patients) were conducted in two more pharmacies (by sixth and last author). Data saturation was then observed. The analysis indicated that emotions were an integrated part of cue orientation and the development of summary codes, why (social) appraisal theory was integrated into the analyses. Appraisal theory highlights how cognition and emotions are interdependent in peoples’ appraisal and reactions to the events in which they take part [20,21]. Hence, the same pharmacy meeting might arouse different feelings in staff and patients and between different staff and patients based on their evaluation of what is going on in the situation. Therefore, apart from registering selected related cues and summary codes as pure cognitive processes, the analysis allowed the identification of emotional responses linked to the selected related cues and summary codes. Besides, it was registered whether an appraisal of a pharmacy meeting leading to an emotional response appeared to be on an individual level or as a member of a social group, i.e., pharmacy staff [21]. Two authors (first and third) carried out this supplementary analysis.

### 2.5. Ethics

Written information about the study was provided to patients and pharmacy staff at the time of recruitment. Oral informed consent from all interviewees was further obtained at the beginning of the interviews. The study was approved by the Danish Data Protection Agency (ref.no. 514-0310/19-3000). The collected data were stored according to the EU rules of GDPR.

## 3. Results

Twelve focus group interviews were conducted during 2017 and 2018. Forty-eight persons participated in the focus groups, including 20 patients, 22 pharmacy staff members and 6 pharmacy researchers. Between 3 and 5 participants were included in each interview. Patient gender was equally represented, but the majority of patients were over 50 years, whereas approximately 2/3 of the staff participants were women and the majority were under 50 years of age. The majority of researchers were women over 50 years of age. The interviews lasted between 55–95 minutes.

Quotes illustrating central elements of the identified ‘*selected related cues*’ and related ‘*summary codes*’ are presented in Table 1, Table 2 and Table 3 for the three different groups. 

### 3.1. Pharmacy Staff

#### 3.1.1. Selected Related Cues

The predominant selected related cue noticed by staff was the (type of) patient. Types of meetings were mentioned by participants in especially one focus group. Further, many communication elements were described, such as language barriers and power balance (please see Table 1).

#### 3.1.2. Summary Codes

Various types of patients were described by staff, including positive patients with a general interest in counselling and an interest in their own health, indifferent patients, busy patients, sensitive patients needing discretion, insecure patients with many questions, and patients who sought information prior to entering the pharmacy. The different types of patients were mainly perceived according to their interest in receiving counselling from staff. This was often described as a personal trait of the patient and, in fewer cases, as a consequence of other demographics, such as age, education, and language. 

Pharmacy staff perceived that the type of patient impacted how the encounter developed. If the patient was interested in a dialogue about medicines, the encounter often developed to address the needs of the patient, whereas a meeting with a patient without interest would be rather short. However, some participants in one pharmacy (pharmacy 2) described how they tried to develop the patients’ interest in counselling by ‘planting seeds’.

A few staff participants, particularly in pharmacy 3, described different types of meetings that differed according to the degree of empathy between the individuals, if the content was technical and if staff managed to interest the patient in counselling. Short meetings (with no communication about medicines) were mentioned in all pharmacies.

Several communication elements that were perceived to influence the meetings were described: expectations, language barriers, a similar notion by both parties of what should take place, the power balance, the importance of getting a good start to the encounter and staff feeling disturbed by phones. A few staff members described how they themselves influenced the encounters. In these cases, the way staff phrased their questions and their body language was perceived to influence the meetings, but a few also added that it depended on whether they had a good day, whether they felt the topic to be embarrassing and whether they were able to read the patient correctly. Variations were seen between the focus groups in the different pharmacies in how many communication issues were discussed, including if pharmacy staff’s own performance was perceived to influence the development of the encounters.

The emotions of staff linked to the described cues and summary codes were in many cases related to the fact that staff felt dependant on the interest of the customer regarding whether or not they could fulfil their perceived social role as medicine counsellors. Hence, different types of patients to a large extent evoked the same types of feelings in staff. For example, those patients being interested in counselling provoked positive emotions in staff whereas those patients refusing counselling evoked negative emotions in terms of disappointment.

### 3.2. Patients

#### 3.2.1. Selected Related Cues

Patients were more varied than staff with regard to predominant selected related cues. Approximately one-third of the patients noticed the type of staff they met at the pharmacy. Another third noticed the content of the meeting. Finally, some patients emphasized the situation ‘around’ the meetings, i.e., if it was a busy pharmacy day and if the requested medicine was in stock. Some patients also described communication elements (please see Table 2).

#### 3.2.2. Summary Codes

Patients who noticed staff discussed whether there was good personal chemistry between them and staff and if they felt pharmacy staff listened to them. Some emphasized that having a personal interaction with staff was important. A few patients said they liked meetings where the two parties had a laugh and did not necessarily talk about the medicine. A central element in the interpretation of staff was whether staff members were explaining too much and/or asking too many questions about the medicine. This element was important because staff were perceived as sometimes overlooking that the patient was not interested in the counselling, and staff thereby displayed a basic lack of interest in the patient. A few patients defended the routine of (over-) informing the patient since this was perceived to be in the best interest of the patient and that staff were only fulfilling their healthcare role.

Other patients focused on the content of the encounter and how it proceeded. Examples of content included how to take the medicine, correct dosage, possible adverse drug events, and drug-drug interactions, and who initiated the discussion about the medicine. 

Some communication elements influencing the meetings were also described; however, fewer than by staff. The aspects mentioned were differences between pharmacies including how the pharmacy owner influenced the atmosphere of the pharmacy also with regard to counselling. Patients who had experienced management responsibilities themselves in their professional career seemed to notice this aspect. A few patients described how they themselves influenced the meeting, for example, by being too quick to judge (negatively) the staff.

Patient ascribed different emotions to pharmacy meetings with some being in general highly content whereas others were more dissatisfied thereby clearly illustrating that patients bear different meanings and emotional responses to the same type of events. There seemed to be a certain pattern between emotions evoked by pharmacy encounters in relation to the cue orientation. Hence, patients who were satisfied with the encounters appeared to focus on the content in their cue orientation, whereas patients who were not always satisfied focused more on the staff ability to create a personal meeting (or the situation around the encounter).

### 3.3. Pharmacy Researchers

#### 3.3.1. Selected Related Cues

The pharmacy researchers described multiple selected related cues, i.e., more and different types of cues than staff and patients. The selected related cues included content of the meetings, the situation around the meeting, the length and outcome of meetings, the type of patient, and various communication elements, for example, how the two parties were influencing each other (please see Table 3).

#### 3.3.2. Summary Codes

The content of a meeting that was influenced by a specific situation was specifically described by the researchers. Examples included an encounter about the medicine being out of stock or discussions about high prices, errors in the prescription, IT problems (electronic prescription not available), etc. 

Hence, researchers, in contrast to staff and patients, gave specific examples of non-medical content in the encounters. The researchers specifically made a distinction between whether the content was about medicine or not, and noted if the staff took the initiative during the meetings to discuss the medicine, and how the patients responded. Further, they described elements such as the length and occurrence of a problem, for example, whether the problem was solved and the satisfaction with the encounter of the involved parties.

Different types of patients were also described by the researchers; for example, some patients were perceived as asking many questions, but patients were also perceived as emotional, complaining, affected by a psychological disease, and getting the medicine for the first time – aspects that in some way influenced the encounters. Researchers also stressed the challenges around obtaining fruitful meetings and gave examples of how the two parties influenced each other, for example, on an emotional level. One example of this was how an unpleasant situation might arise because the two parties repeatedly reacted towards the other person’s stress. The staff skills to cope with such situations were discussed. Otherwise, the researchers did not distinguish between types of staff as much as between types of patients. In general, as part of the construction of summary codes, the researchers explicitly described emotional aspects as an integral part of prescription encounters as compared to staff and patients who were influenced by emotions but did not describe this specifically.

Researchers were in general reluctant to describe what they perceived to be the optimal pharmacy encounter hence were hesitant to display their own emotions towards pharmacy meetings. However, some criticism or dissatisfaction of pharmacy staff was shown with regard to them not always being able to assess adequately the needs of the patient. Researchers appeared aligned in their deliberate lack of emotional response to pharmacy encounters perhaps thinking that displaying emotions is not an acceptable social role of a pharmacy researcher or that they are looking at the encounters from a distance and are hence not personally involved.

## 4. Discussion 

Many and different types of selected related cues in prescription encounters were described by pharmacy staff, patients and researchers. The three groups noticed different cues. Staff particularly noticed patients whereas patients were more divided and grouped into three overall cues, including staff, medical content, and the situation around the encounter. Pharmacy researchers noticed multiple cues. Different elements including emotions were included in the construction of the cues. For example, patients’ interest in talking about the medicine or not was a central element in staff’s perception of ‘types of patients’ and influenced their emotional response towards the encounters, whereas staff skills to create a personal meeting were included in patients’ descriptions of and satisfaction with ‘types of staff’. Pharmacy staff and pharmacy researchers’ emotional responses involved in their perception of pharmacy encounters were more univocal than those of patients. 

### 4.1. Differences in Cue Orientation between Patients and Staff

Considerable difference as described above was found in the selected related cues noticed by staff and patients and the meaning (summary codes) they ascribe to them. To our knowledge, this is the first study to focus on cue and cue differentiation between community pharmacy staff and patients; however, other researchers have previously identified differences between staff and patients, for example, with regard to preferences of the roles of the community pharmacist [22,23]. In these studies, staff more than patients was shown to think that they should be involved in detecting and solving patients’ drug-related problems. Such results were partly observed in our study, in particular, in relation to some patients feeling content by having a casual talk rather than a talk about medicines; in contrast to staff who focused on whether a patient was interested in a dialogue about medicine or not. Hence, staff and (some) patients do not agree on the items they appreciate being discussed in prescription encounters. Yet patients in this study appeared to differ in this view as we also identified a group of patients who were satisfied with staff asking questions about their use of medicines. 

That staff primarily focused on types of patients and how they influenced the encounter, which might be explained by staff aiming at fulfilling their perceived social role as medical counsellors. Hence, pharmacists in many countries base their counselling on the individual patients, and trying to ‘read’ the patient is perceived as an integral part of good communication skills [24].

A reason for the identified differences in cue orientation between patients and staff could be that when relationships do not work optimally the involved parties focus more on the underlying relationship than the content [15]. In fact, the predominant type of cue of the majority of staff and some patients pertained to the relationship i.e., a focus on the opposite person and how this person influenced the meeting. In supplement, the patients who appeared satisfied with the counter meetings focused primarily on the content. This result thus indicates that both patients and staff are dissatisfied with prescription counselling today, yet for different reasons. Hence, the dissatisfaction is shown by the aspects/cues they notice around pharmacy encounters and the meaning built into them. Assa-Eley at al. (2005) noted the fact that disagreement between patients and staff of what ought to take place during the encounters often goes unrecognized [23], which might explain why unsatisfactory practices are being repeated.

### 4.2. Differences in Cue Orientation between Patients

We observed two overall types of patients: those who focused on the type of staff and those who focused on content of the dialogue. A few patients noticed the situation around the encounter.

Renberg et al. (2011) investigated how patients differed according to their ideal prescription pharmacy encounter and found two overall groups: those who focused on the ‘drug product’ and those who focused on ‘personal support’ from staff [25]. Each overall group consisted of sub-groups/factors. For example, ‘personal support’ included both the sub-group IV emphasizing a competent pharmacist who should offer individual advice since the patient usually does not like to make health decisions alone, as well as sub-group V emphasizing privacy and personal contact as some patients are not fond of a traditional professional-client relationship [25].

When comparing our results to those of Renberg, the two predominant patient groups in this study both pertain to the ‘personal support’ group but in different ways representing both sub-group IV and V. Hence, in contrast to the study of Renberg, only a few of the patients in this study described the pharmacy as a place that is only relevant for them in order to pick up medicine. However, this type of patient was described by staff in our study, which might have to do with how patients were recruited and the willingness of patients to participate. The study by Renberg was conducted using Q-methodology with a more overall perspective on pharmacy practice. Also, the study was conducted in 2008 (in Sweden), which might explain some of the differences. Further, community pharmacies worldwide have, for the last decade, attempted to embrace an expanded role in patient-centred counselling, which some patients may now have started noticing and appreciating. 

### 4.3. Practice Implications

Researchers described many types of selected related cues compared to pharmacy patients and staff, which might be considered both appropriate and explainable considering the complexity of the field [26]. Hence, researchers in the field of pharmacy communication seem to be open to studying many different elements related to pharmacy encounters. However, for future pharmacy practice research aiming to improve counselling, researchers and educators should consider paying special attention to the issues of importance to staff and patients. 

Another important aspect relates to the staff’s focus on the types of patients, i.e., the majority of staff seem to perceive that the way the encounter develops is influenced by the type of patient and not by themselves. This perception affected staff emotionally. Rather than trying to change the situation many staff members appeared to accept it. Appraisal theory points to the phenomenon of applying relevant adaptive emotions, i.e., a situation where emotions are selected based on the individual’s evaluation of the situation comparing the capabilities and resources of the individual with the requirements of the situation [20]. Hence, on one side disappointment as the emotional response to lack of patient interest in pharmacy counselling is relevant in the sense that it conserves resources in a situation that cannot be changed. However, this emotional response is at the same time unfortunate since staff then prevents themselves from trying to develop the encounters further. Hence, being aware of emotional responses in relation to pharmacy encounters might be one way of furthering communication at the counter for pharmacy staff.

An indication of lack of satisfaction with prescription encounters of both patients and staff was identified. Patients, through their descriptions of prescription encounters, displayed the aspects they appreciate and often lack in the encounters, in particular, more personal contact. As most staff participants didn’t describe this aspect, a particular attention by staff in this area is warranted to improve counselling. 

### 4.4. Limitations

The results of this study might be influenced by selection bias. Contact persons in the five pharmacies responsible for recruiting staff and patients might have (subconsciously) selected certain kinds of participants, for example, those who are more in favour of pharmacy counselling. Social desirability might also have played a role as the interviews with patients and staff took place in the pharmacy. However, a group of patients who were not overly satisfied with pharmacy encounters was also included in the sample.

Some of the discrepancies between staff and patients with regard to cue orientation might be explained by the fact that the majority of patients were older than 50 years, whereas the majority of staff were younger than 50 years. As cues and summary codes are influenced by both individual and collective experiences, age is expected to play a role in forming expectations.

The enrolled researchers reflected on different kinds of experiences when contemplating pharmacy encounters. Some of the researchers reflected on experiences as former community pharmacists, and others reflected on their experiences as patients or insights gained through their research. These different approaches might explain why researchers were particularly varied in their cue orientation. That researchers noticed more selected related cues than staff and patients also illustrates how selective the last two groups are in their cue orientation and, thus, how the theoretical approach of cue orientation is useful in investigating differences in cognitive (and emotional) elements constituted in underlying perceptions of pharmacy encounters. 

The perceptions and views of pharmacy encounters described by interviewees do not necessarily reflect how the pharmacy counter meetings take place. To verify how meetings actually occur, observational studies are needed. However, identified cues and summary codes are important tools when trying to understand why pharmacy encounters today are not perceived as satisfactory by all involved parties. This knowledge might be used as a basis for community pharmacy staff to further develop their communication skills.

## 5. Conclusions

Cue orientation, summary codes, and related emotions in pharmacy prescription encounters differ considerably between community pharmacy staff, patients, and pharmacy practice researchers. These differences could constitute some of the identified challenges in pharmacy counselling today. For example, some patients noticed aspects around a personal encounter whereas staff appeared more oriented towards discussing the patient’s medicine. A group of patients, however, was content with the current counselling practices of staff.

## Figures and Tables

**Table 1 pharmacy-07-00084-t001:** Examples of elements (transcript data) built into the summary codes of the selected related cues noticed by pharmacy staff.

Selected Related Cues	Elements in Summary Code
Type of patient	‘I have counseled those [ed. patients] who are open, that like to know more and who seek insight into their own disease. That’s a good dialogue, I would say.’ (Pharmacy 1)
	‘And then I have written a regular customer who is used to coming here and they are used to us helping them and it is at our place that they look for advice and confirmation—and we have to help them with all kind of things related to their treatment.’ (Pharmacy 1)
	‘And then there are some who are a bit defiant who don’t want to listen because the doctor knows best and we should not interfere…In those cases, I always try to plant a seed, so perhaps they can think about it and then come back.’ (Pharmacy 2)
	‘I experience that there is a bit of difference between customers, sometimes we have new customers who are not well informed, who should have some counseling, in contrast to people who have had it [ed. the medicine] for 20 years. There is a difference if they really want to listen.’ (Pharmacy 4)
	‘There is a big difference if you are speaking to a man or a woman because women like to share and men don’t.’ (Pharmacy 4)
Types of meetings	‘I think we have a lot of what I have called ‘the intimate meeting’, where the customer opens up, where we get to talk about it, where we are allowed to get under their skin and become intimate in our talk.’ (Pharmacy 3)
	‘I have divided into a professional academic meeting where you are allowed to bring some information to the customer which the customer was not aware of.’ (Pharmacy 3)
	‘I have divided them into quick encounters without information. The customers knows everything or they had the medicines for many years and think they know everything about it… And then we have the in-depth encounters where you take into consideration that the customer is a new user of the medicine and where the customer is interested in receiving information.’ (Pharmacy 5)
Communication elements	‘Cultures, high/ low status, habits of informing, those who know better, the busy ones, it all have an influence on us in the interaction. It can be noise, a printer which is noisy, a telephone that rings, somebody who wants you to answer the phone or who wants to ask you something: ‘where can I find this product?’ There are so many things that influence the encounter which makes it different for the customer.’ (Pharmacy 1)
	‘And I have written the interactions in which the customer comes in with certain expectations and then meets something else – and that can go both ways: ‘Thank you so much’ or ‘Can’t we finish so I can get out of here?’’ (Pharmacy 2)
	‘I agree that if you have a bad start of the encounter then it can influence the rest of the meeting with this customer.’ (Pharmacy 3)
	‘Person 2: Language barriers, I think we experience that with every second customer Person 4: And it gives, as you said, impatient customers because if we have the language barriers then it takes time.’ (Pharmacy 4)

**Table 2 pharmacy-07-00084-t002:** Examples of elements (transcript data) built into the summary codes of different selected related cues noticed by patients.

Selected Related Cues	Elements in Summary Code
Type of staff	‘Yes, then you get one of the staff with whom you have special interaction.’ (Pharmacy 1)
	‘I don’t mind [ed. getting information] but when I say: “Yes I’ve taken it for 25 years” they say: “Yes, but you have to be aware of…”—“But try to listen to what I’m telling you. I know it. I discuss it with my doctor” and then they say: “Yes, yes but…” and they keep on.’ (Pharmacy 1)
	‘I just want more personal contact, something more personal than a conversation only about the medicine or how expensive it is.’ (Pharmacy 5)
Content of meeting	‘They ask whether you have any questions regarding the medicine, if the medicine is new to you or if you have any side effects, you would like to discuss.’ (Pharmacy 2)
	‘I have experienced very professional encounters…It’s: “Do you know how to take it?” and the dosage they focus on.’ (Pharmacy 4)
	‘With regard to the counseling, I often feel that they advise you on how many tablets to take and how often.’ (Pharmacy 5)
Situation around the meeting	‘I think it’s quite nice to come to Copenhagen because they always have the products I need.’ (Pharmacy 3)
	‘You could observe that there is some kind of stress from their (ed. the staff) side. I think some times that if there is a long queue waiting then they perceive that it should be a bit quicker.’ (Pharmacy 4)
	‘When they are very busy then they try to make it very short and concise – and then it’s out with this guy and in with the next.’ (Pharmacy 5)
Communication elements	‘I have lived in many different places and been a customer in many pharmacies. You sense if there is a good spirit in the pharmacy. I haven’t yet sensed the spirit of the new owner down here.’ (Pharmacy 1)
	‘Interviewer: have you ever experienced to be positively surprised?K3: Oh yes. It concerns all aspects of life—also down here. Oh no not him and then it turns out well. And then it’s a good experienceK2: basically it’s because your first judgment is wrong.’ (Pharmacy 1)
	‘And it might be that there is another tone here in the counseling area after the new owner has started…You feel it when you enter….It has a positive influence on the staff.’ (Pharmacy 2)
	‘I feel the quality can be different from pharmacy to pharmacy. It can be very different…depending if you are in a big pharmacy or in a small branch and who is behind the counter.’ (Pharmacy 4)

**Table 3 pharmacy-07-00084-t003:** Examples of elements (transcript data) built into the summary codes of different selected related cues noticed by pharmacy researchers.

Selected Related Cues	Elements in Summary Code
Content	‘I have myself experienced what could be defined as generic substitution where we discuss the price, the package, the looks, the drug and I have a lot of different experiences with that.’ (University 1)
Situation around meeting (influencing content)	‘And then there are the problems when the doctor hasn’t sent the prescription and you [ed. being the staff] talk a lot about that, problems with the doctor sending the wrong medicine.’ (University 1)
	‘And the problems can be related to the people or due to IT-problems, it can be the prescription-server, it can be something with the IT that doesn’t work, it can be drug-shortages…’ (Pharmacy 1)
	‘And then there are very practical matters such as the customer complains because there has been a mistake, or the customer can’t get the medicine due to drug shortages, the customers finds the medicine expensive, the customer doesn’t speak Danish. So there are a lot of meetings being event-dependent.’ (University 2)
Type of patient	‘And then I thought about some of the customers who are very worried about something or that you received some new medicine or that you received a new diagnosis.’ (University 1)
	‘There are many different kinds of meetings depending on what type of customer you have, and that’s the way it should be.’ (University 2)
	‘…the customer is in a hurry, the customer doesn’t want to talk about something because it’s a taboo, that the customer is emotionally affected…’ (University 2)
Communication elements (including interaction and emotions)	‘And then we have customers with misuse problems and the pharmacist knows this, and then the customer thinks that the staff member looks at him in a strange way and perhaps the pharmacist does, but even so, the pharmacist doesn’t, you feel awkward as the customer. But also, where the pharmacist doesn’t dare or care to make the effort because it’s unpleasant.’ (Pharmacy 1)
	‘I think in most cases that it is the customer who is a bit aggressive. They have been waiting in the queue for a long time, they are aggressive, then you [ed. being the staff] turn a bit aggressive because the other party, whoever that is, is influenced by it.’ (University 1)
	’…where the pharmacist or the pharmacy technician invites you for a talk about the treatment or the drug counseling. Where the patient accepts – and other scenarios where the pharmacist or the pharmacy technician invites for talk about drug advice where the patient declines.’ (University 2)
	‘There can be different parameters which have an influence such as age, language – is there a language barrier which you often experience in the pharmacies and how it influences the meeting.’ (Pharmacy 2)

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
