# Peer review of "Patients’, Pharmacy Staff Members’, and Pharmacy Researchers’ Perceptions of Central Elements in Prescription Encounters at the Pharmacy Counter"

_pharmacy, 2019, doi:10.3390/pharmacy7030084_

Round 1

Reviewer 1 Report

I would suggest to replace the term "coustomers" with "patients". 

Author Response

I would suggest to replace the term "customers" with "patients".

“customers” have been replaced by “patients”

Reviewer 2 Report

Study objectives: The study objectives were not clear to me, specifically how identified/investigated  'cues' in prescription encounter will impact the way pharmacy counselling take place.

Results: The results section could have been better organized. Table 1 was very confusing. It would be more clearer if the table was broken into three tables (one per each study group- staff table, customers table and researchers table). Further, in each table, identified cues would be listed in addition to summary codes. 

Discussion: Section 4.1 only commented on staff focus and did not highlight what differences exist in cue orientation between customers and staff 

Conclusion: the conclusion section is very brief and the message was not clear and different from the abstract conclusion section

Author Response

Study objectives: The study objectives were not clear to me, specifically how identified/investigated 'cues' in prescription encounter will impact the way pharmacy counselling take place.

a part of the introduction has been re-written to better illustrate the importance of investigating cues

Results: The results section could have been better organized. Table 1 was very confusing. It would be more clearer if the table was broken into three tables (one per each study group- staff table, customers table and researchers table). Further, in each table, identified cues would be listed in addition to summary codes.

table 1 has been broken into 3 tables as suggested including listing the matching cue

Discussion: Section 4.1 only commented on staff focus and did not highlight what differences exist in cue orientation between customers and staff

à The sections has been revised to better illustrate that the section regards differences in cue orientation

Conclusion: the conclusion section is very brief and the message was not clear and different from the abstract conclusion section

the conclusion in the manuscript has been expanded

Author Response

The goal of this research is to understand the basis of current challenges in pharmacy counseling by investigating cues in the prescription encounter—both customer’s and staff members interpretation of various cues. This is an interesting and important contribution to the literature. That said, I recommend revising this manuscript to enhance the methodology, reporting of results and discussion.

Materials and Method: I would like to see a more detailed description of social perception and appraisal theory as it relates to your study, especially since it speaks directly to role identity, a term that isn’t mentioned anywhere in your article, but is especially relevant. I realize you may be condensing due to space limitations, but there is much more to this theory that could help unpack your results and implications, so I recommend expanding this section.

please see below

I’m a bit confused as to the inclusion of the pharmacy researchers’ cues to explore investigator bias. While it is certainly appropriate (and expected) that the researchers will address any bias in their analysis of the results, I don’t see value-added in including an analysis of your own thoughts regarding pharmacy/customer encounters and recommend omitting this entirely from the manuscript. Simply address the potential bias that could have colored your interpretation of the results in the limitations.

àAs the two other reviewers did not raise this issue we have decided to keep the results about researchers’ cue orientation. To fulfill the request of the reviewer we have however tried to frame the researchers’ perspective in a hopefully more clear way, and also described more in detail what is the added value of this data in the discussion. It seems that researchers base their understanding of pharmacy encounters based on other cues than the parties involved in the communication (patients/pharmacy staff), and hence the risk that researchers are unaware of this when doing research, we think is worthwhile to illustrate. 

The second question posed to focus group participants concerned their opinions about the role of community pharmacies in society (line 96-97). It is unclear how this question would help you learn about customers’ and pharmacists’ perceptions of encounters.

the reason behind this question is now made clearer in the method section

I’d like to see greater detail regarding the data analysis process. Consider that methodology sections should be written so the study could be reproducible.

à we have included more details about the data analysis process

Results and Discussion: I encourage the authors to revisit the data to seek out more nuance in the responses. Consider your analysis in terms of the theoretical framework and the challenges you mentioned in the introduction as a way to further make sense of the data. Similarly, the results should link back to social perception and appraisal theory and also speak to the challenges identified between pharmacists and customers at the point of counseling. What do your findings offer in terms of greater understanding of pharmacist/customer interactions and recommendations for how to improve?

as our results showed that emotions are interlinked with cues and summary codes we have integrated (social perception and) appraisal theory to in particular illustrate feelings in relation to pharmacy encounters better. Hence, we have described this added theoretical part shortly in the analysis part of the method section. We have also added a paragraph to the result section describing the identified feelings of each of the three groups, and whether the feelings appear to be on an individual or social group level. This is also  further commented on in the discussion  

I hope the authors consider revising this manuscript and resubmitting as it has the potential to make a unique and timely contribution to the literature.

Round 2

Reviewer 3 Report

The authors revisions addressed my concerns with this manuscript.